# Vertical-Longitudinal Coupling Effect Investigation and System Optimization for a Suspension-In-Wheel-Motor System in Electric Vehicle Applications

Ze Zhao [1], Lei Zhang [1,*], Jianyang Wu [2], Liang Gu [1] and Shaohua Li [3]

1   National Engineering Research Center for Electric Vehicles, Beijing Institute of Technology, Beijing 100081, China
2   Beijing Institute of Space Launch Technology, Beijing 100076, China
3   State Key Laboratory of Mechanical Behavior and System Safety of Traffic Engineering Structures, Shijiazhuang Tiedao University, Shijiazhuang 050043, China
*   Correspondence: lei_zhang@bit.edu.cn

**Abstract:** In-wheel-motor-drive electric vehicles have attracted enormous attention due to its potentials of improving vehicle performance and safety. Road surface roughness results in forced vibration of in-wheel-motor (IWM) and thus aggravates the unbalanced electric magnetic force (UEMF) between its rotor and stator. This can further compromise vertical and longitudinal vehicle dynamics. This paper presents a comprehensive study to reveal the coupled vertical–longitudinal effect on suspension-in-wheel-motor systems (SIWMS) along with a viable optimization procedure to improve ride comfort and handling performance. First, a UEMF model is established to analyze the mechanical–electrical–magnetic coupling relationship inside an IWM. Then a road–tire–ring force (RTR) model that can capture the transient tire–road contact patch and tire belt deformation is established to accurately describe the road–tire and tire–rotor forces. The UEMF and the RTRF model are incorporated into the quarter-SIWMS model to investigate the coupled vertical–longitudinal vehicle dynamics. Through simulation studies, a comprehensive evaluation system is put forward to quantitatively assess the effects during braking maneuvers under various road conditions. The key parameters of the SIWMS are optimized via a multi-optimization method to reduce the adverse impact of UEMF. Finally, the multi-optimization method is validated in a virtual prototype which contains a high-fidelity multi-body model. The results show that the longitudinal acceleration fluctuation rate and the slip ratio signal-to-noise ratio are reduced by 5.07% and 6.13%, respectively, while the UEMF in the vertical and longitudinal directions varies from 22.2% to 34.7%, respectively, and is reduced after optimization. Thus, the negative coupling effects of UEMF are minimized while improving the ride comfort and handling performance.

**Keywords:** in-wheel-motor; unbalanced electric magnetic force; vertical–longitudinal dynamics; road–tire–rotor force; multi-optimization method; virtual prototype

## 1. Introduction

Automotive electrification is rapidly expanding worldwide to tackle the formidable challenges of greenhouse gas emissions and fossil oil depletion. In-wheel-motor-drive electric vehicles (IWMD EVs) employ four in-wheel-motors (IWMs) installed inside each wheel hub to realize direct propulsion [1–3]. This results in better sprung mass packaging flexibility and can potentially enhance vehicle dynamics stability by independently controlling each IWM [4–6]. Switched Reluctance Motor (SRM) is a reasonable choice for IWM due to its high starting torque, wide speed range and high efficiency [1,7]. However, the motor drive and the suspension system are rigidly connected. Under road roughness excitation, an unbalanced electric magnetic force (UEMF) is generated inside the motor

due to the magnetic gap eccentricity [8]. These form a typical closed-loop mechanical–electrical–magnetic coupling system. The eccentricity caused by the mass imbalance of the motor and by external forces significantly contributes to the UEMF, thus further distorting the magnetic gap distribution and aggravating the amplitude of the electromagnetic excitation [9,10].

Numerous studies have been conducted to mitigate the adverse influence of UEMF on vehicle dynamics. These can be generally grouped into the two categories according to vertical and longitudinal dynamics. For the former, the main approach to reducing vertical UEMF is design optimization for either motor or suspension system. The structure of in-wheel motors can be optimized by altering the rotor and stator geometries or the windings to minimize UEMF [11]. In this regard, Wang et al. [12] investigated the influence of the representative parameters of an in-wheel motor on vehicle vertical dynamics. Similarly, Li et al. [13] introduced parallel paths into the windings to reduce the influence of UEMF on the vibration of the motor. However, these approaches fail to take into account road roughness excitation, and merely optimize the electromagnetic characteristics of the motor. Another method to reduce the vertical UEMF is to optimize suspension design in combination of developing enabling control algorithms. The novel dynamic vibration absorbing structure (DVAS) exemplifies the effort [14], which is installed between the motor and the suspension system to absorb the motor-caused vibration. Previous studies have shown that DVAS can effectively suppress the motor-caused vibration when the spring dampers are properly parameterized [15,16]. However, the proposed integration structure is limited by its complexity, and its efficacy has not been fully validated. On the other hand, various suspension control algorithms have been developed to negate the vertical vibration induced by UEMF. This considers the suspension and in-wheel motor as a complete system and uses active suspension control algorithms to reduce the adverse impact of the increased unsprung mass. The common algorithms include fuzzy logic control [17], ceiling damping control [18], optimal sliding mode control [19], $H_\infty$ control [20], and the like. Nevertheless, the existing studies have invariably neglected the impact of the electromagnetic field, which means that the mechanical–electrical–magnetic coupling effect cannot be fully accounted for.

Axle bows, bearing tolerance and longitudinal velocity variation can lead to the longitudinal eccentricity of the IWM, and the resulting UEMF would further cause the longitudinal vibration[21]. Longitudinal vibration that interacts with the complex elastic structures such as tire may significantly compromise the vehicle traction/braking control performance [22,23]. The UEMF-induced longitudinal vibration can be suppressed via structure optimization and traction/braking force control. For instance, Kambe et al. [24] employed a shock absorber to reduce longitudinal vibration. Similarly, Zuo et al. [25] adopted a flexible connection between the rim and the in-wheel motor to modify the inherent characteristics of electric wheels, resulting in decreased longitudinal vibration. Traction or braking forces can be adjusted through sliding mode control [26], $H_2/H_\infty$ control [27], and direct yaw control [28] in order to mitigate the longitudinal vibration. The existing studies on longitudinal vibration mitigation mainly concentrate on regulating torque ripples, and rarely consider the impact of UEMF. In addition, there is lack of a comprehensive evaluation system that can precisely characterize the longitudinal vibration.

However, the vertical and longitudinal vehicle dynamics are always coupled with each other [29,30]. Several studies have been conducted to reveal the vertical–longitudinal coupling effects for IWMD EVs. For instance, Qin et al. [31] formulated a longitudinal–vertical quarter-vehicle model and analyzed the impact of a random and constant UEMF on vertical and longitudinal vehicle motions. However, this study mainly focused on the influence of the vertical UEMF on longitudinal vehicle motion by assuming a constant longitudinal eccentricity. Chao et al. [32] built a quarter-vehicle longitudinal–vertical coupling model to improve the comprehensive performance of the electro-mechanical braking system. In their study, the evaluation metrics were mainly related to energy efficiency and longitudinal characteristics, neglecting the vertical performance. Ricciardi et al. [33] presented an integrated dynamic control for wheel torque distribution in the longitudinal

direction and for vehicle suspension force in the vertical direction. Their simulation results indicated that this so-called "ride-blending" control can reduce the pitch response of the vehicle body while keeping the dynamic tire force within the safety constraints. However, this study ignored the electro-mechanical coupling effect of IWM, and did not consider the effect of UEMF on vehicle dynamics. Thereby, there is a need to investigate the coupling mechanism of vertial and longitudinal vehicle dynamics caused by UEMF, and their relevant alleviation methods.

Generally, simplified tire models only including spring characteristics are used in vehicle control; however, the increase in unsprung mass and the high-frequency excitation generated by the motor directly affect the working conditions of tires [34]. Typical tire models include physical and empirical models. The brush and ring models are the two main forms of physical models, while the well-known magic formula can be considered as an empirical model. In the existing studies, the Brush and Magic Formula tire models have been used to simulate the tangential tire force characteristics [35,36]. The ring model (RRM) and the flexible ring model (FRM) are commonly used to describe vertical tire dynamics [37,38]. In the RRM, the residual stiffness is introduced between the contact patch and the rigid ring to represent the static tire stiffness in the vertical direction. However, as the tire belt deformation is not considered, this model is only suitable in the lower-frequency ranges [37]. Meanwhile, the FRM employs a large number of segments interconnected by springs and dampers. Its bandwidth is beyond the frequency range of 150 Hz; however, it can occasionally describe the low-frequency characteristics [38]. As for IWMD EVs, its vertical motion is exposed to both high- and low-frequency excitations [22]. Consequently, in order to comprehensively analyze the vertical–longitudinal coupling characteristics of the road–tire–IWM system, it is necessary to integrate the above-mentioned tire models for better representation.

As described above, there are two obvious vertical–longitudinal coupling systems existing in IWMD EVs. These are the electromechanical–magnetic coupling system and the road–tire–IWM system; the UEMF and the road–tire–rotor force (RTRF) are their respective characteristic forces. Therefore, it is necessary to build an accurate and comprehensive model to investigate the influence of these two systems on vehicle vertical–longitudinal dynamics. The exclusive contributions of this study are summarized as follows.

1.  A UEMF model is established and used as an internal excitation of the suspension-in-wheel-motor system (SIWMS). The electro-mechanical mechanism of UEMF and its effect on vehicle vertical–longitudinal dynamics are comprehensively studied under various road conditions.
2.  A RTRF model that can capture the transient tire–road contact patch and tire belt deformation is proposed to accurately describe the vertical–longitudinal coupling effect of the road–tire–motor system. The RTRF model is then incorporated into the SIWMS model for better modelling accuracy.
3.  A comprehensive evaluation system is proposed to describe the vertical–longitudinal dynamics of IWMD EVs. The ride comfort and handling performance are improved by optimizing the key parameters of the SIWMS.

The remainder of this paper is organized as follows. Section 2 introduces the quarter-SIWMS model by incorporating the UEMF and the RTRF model. Section 3 provides the verification of the SIWMS model in a virtual prototype by establishing a high-fidelity multi-body vehicle model. Section 4 elaborates on the UEMF effects on the coupled vertical and longitudinal vehicle motions and presents the details on parameters optimization of the SIWMS. Finally, our key conclusions are summarized in Section 5.

## 2. The Quarter-SIWMS Model

The primary objective of this study is to fully understand the electro-mechanical-magnetic coupling effects of IWMD EVs. Hence, a precise vehicle dynamics model is established to reveal the contributions of different factors. In this study, the quarter-SIWMS

model consists of three sub-models: (1) the excitation source model, (2) the RTRF model, and (3) the nonlinear three-directional coupled (NTDC) model.

### 2.1. The Excitation Source Model

The SIWMS model is characterized by the internal and the external excitation. The internal excitation is the investigated UEMF, while the external excitation is mainly ascribed to the road roughness.

### 2.1.1. The UEMF Model

In this study, a 5-kW exterior rotor SRM with 8/6-four phases [39] is selected as the IWM IWMD EVs. For an SRM, the radial and tangential components of the electromagnetic force depend on the radial and tangential components of the flux density [40]. As the magnetic permeability of the air gap is much lower than that of the ferromagnetic material, the radial component of the flux density is greater than the tangential [41,42]. This is verified by existing studies, which have shown that the radial force is about 16.6 times more than the tangential force in an SRM [43]. This means the rotor and stator eccentricity can cause a large radial force, leading to severe vibration issue. The UEMF is produced by the coupled effects of the electromagnetic and mechanical fields, and represents the resultant global magnetic force acting on the rotor and stator due to the asymmetric magnetic field distribution in the air gap. The UEMF generation process is shown in Figure 1, where $F_e$ is the electromagnetic force, $F_t$ is the tangential force and $F_r$ is the radial force.

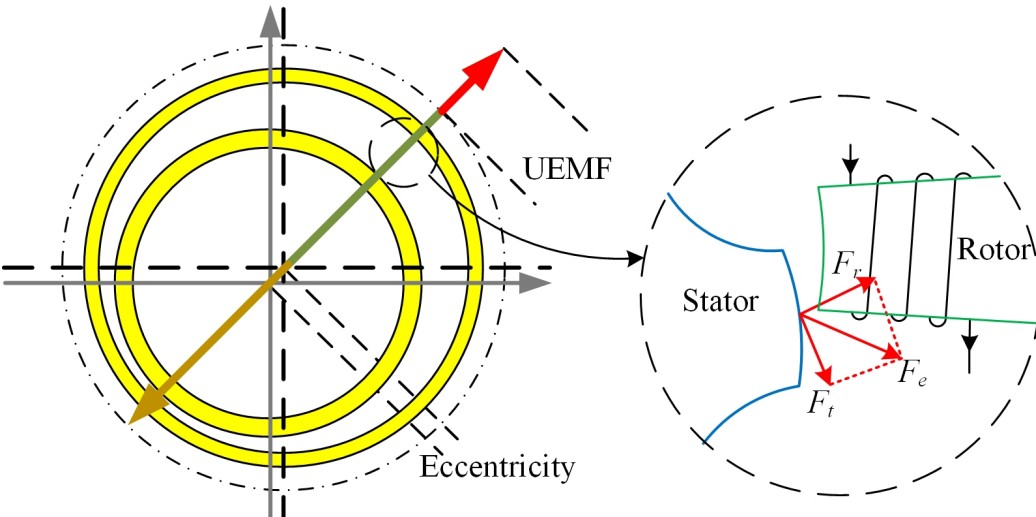

**Figure 1.** The generated UEMF in an SRM.

For SRMs, the magnetic co-energy $W(i, \theta)$ is determined according to the current $i$ and the phase inductance $L(\theta, i)$, where $\theta$ is the rotor angle. The first three terms of the Fourier expansion are given by

$$L(\theta, i) = L_0(i) + L_1(i)\cos(N_r\theta + \pi) + L_2(i)\cos(2N_r\theta + 2\pi) \tag{1}$$

where $L_0$, $L_1$, and $L_2$ are calculated by

$$\begin{cases} L_0(i) = \frac{1}{2}\left[\frac{1}{2}(L_a(i) + L_u) + L_m(i)\right] \\ L_1(i) = \frac{1}{2}(L_a(i) - L_u) \\ L_2(i) = \frac{1}{2}\left[\frac{1}{2}(L_a(i) + L_u) - L_m(i)\right] \end{cases} \tag{2}$$

where $L_a$, $L_u$, and $L_m$ are the inductances at the fully aligned ($\theta = 30°$), unaligned ($\theta = 0°$), and intermediate positions, respectively. These parameters can be fitted with polynomials

based on finite element or experimental analysis. Considering the relationship between the flux and the inductance, the $k$-th phase flux linkage can be derived as

$$
\begin{aligned}
\psi(\theta, i_k) &= \int_0^{i_k} L(i_k, \theta)\mathrm{d}i_k \\
&= \frac{1}{2}\left[\cos^2(N_r\theta) - \cos(N_r\theta)\right]\sum_{n=0}^{N} c_n i^n \\
&\quad + \sin^2(N_r\theta)\sum_{n=0}^{N} d_n i^n \\
&\quad + \frac{1}{2}L_u i_k\left[\cos^2(N_r\theta) + \cos(N_r\theta)\right]
\end{aligned}
\tag{3}
$$

where $\psi$ is the flux linkage, $N_r$ is the number of salient poles in the rotor, $c_n = a_n - 1/n$ and $d_n = b_n - 1/n$. According to the Faraday's law, the phase voltage can be derived by

$$
U_k = R_k i_k + \frac{\mathrm{d}\psi_k}{\mathrm{d}t} = R_k i_k + L_k(\theta, i_k)\frac{\mathrm{d}i_k}{\mathrm{d}t} + \omega\frac{\partial\psi_k}{\partial\theta}
\tag{4}
$$

where $\omega$ is the rotational speed of the rotor. The phase current can be given by

$$
i_k = \int \frac{U_k - R_k i_k - \omega\frac{\partial\psi_k}{\partial\theta}}{L_k(\theta, i_k)}\mathrm{d}t
\tag{5}
$$

For the constant phase current $i$, the relationships between the magnetic co-energy $W(i, \theta)$, torque $T$, and radial force $F_r$ can be obtained by

$$
T = \left.\frac{\partial W(\theta, i)}{\partial\theta}\right|_{i=\mathrm{const}}, F_r = \left.\frac{\partial W(\theta, i)}{\partial l_g}\right|_{i=\mathrm{const}}
\tag{6}
$$

where $l_g$ is the air gap between the rotor and the stator. The phase torque can be deduced by

$$
\begin{aligned}
T_k &= \left.\frac{\partial W(\theta, i)}{\partial\theta}\right|_{i=\mathrm{const}} = \int_0^{i_k}\frac{\partial\psi(\theta, i_k)}{\partial\theta}\mathrm{d}i_k \\
&= \sin(N_r\theta)\sum_{n=1}^{N}\frac{1}{n}e_{n-1}i_k^n + \sin(2N_r\theta)\sum_{n=1}^{N}\frac{1}{n}f_{n-1}i_k^{n_k}
\end{aligned}
\tag{7}
$$

where both $e$ and $f$ are the intermediate functions. The former, $e$, can be obtained by $e_n = (1/2)N_r c_n$ with $e_1 = (1/2)N_r(c_1 - L_u)$. The latter, $f$, can be calculated by $f_n = N_r d_n - e_n$ with $f_0 = 0$ and $f_1 = (1/2)N_r(2d_1 - c_1 - L_u)$.

Based on the phase torque $T_k$, the radial force of the $k$th phase can be calculated by [44]

$$
F_{rk} = -\frac{T_k\delta}{l_g}
\tag{8}
$$

where $\delta$ is the overlap between the rotor and the stator. The presence of the non-zero $l_g$, known as eccentricity, results in UEMF. The UEMF is defined as a difference in the radial forces between the two opposing stator poles. There are many reasons for eccentricity, such as poor manufacturing accuracy or dynamic coupling effects [45]. The air gap eccentricity $l_g$ can be decomposed into two resultant components denoted as $\varepsilon_x$ and $\varepsilon_z$. In this study, the air gap in the $x$-direction that is caused by axle bows, bearing tolerance and longitudinal road inputs is denoted as $\varepsilon_x$. The dynamic eccentricity in the $z$-direction, defined as $\varepsilon_z$, describes the dynamic coupling effect of the suspension, IWM, and road roughness. The relationship between the UEMF, its components (i.e., $F_{uv}$ and $F_{ul}$), and mixed eccentricity is depicted in Figure 2.

Based on the definitions of UEMF and mixed eccentricity, the vertical UEMF $F_{uv}$ can be calculated as [14]

$$
F_{uv} = \sum_{k=1}^{4}\left[\left(-\frac{T_k\delta}{l_g - \varepsilon_y\cos\beta_k} + \frac{T_k\delta}{l_g + \varepsilon_y\cos\beta_k}\right)\cos\beta_k\right]
\tag{9}
$$

Similarly, the longitudinal UEMF $F_{ul}$ is provided by

$$F_{ul} = \sum_{k=1}^{4} \left[ \left( \frac{T_k \delta}{l_g - \varepsilon_x \sin \beta_k} - \frac{T_k \delta}{l_g + \varepsilon_x \sin \beta_k} \right) \sin \beta_k \right] \tag{10}$$

where $\beta$ is the phase structure angle ($\beta_1 = 0°$, $\beta_2 = 45°$, $\beta_3 = 90°$ and $\beta_4 = 135°$) and the nominal air gap is 0.8 mm.

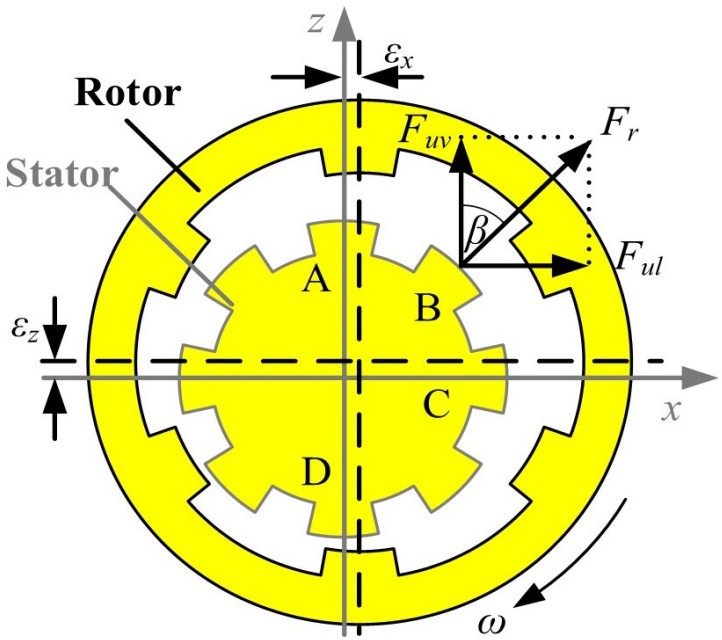

**Figure 2.** The vertical and longitudinal UEMFs induced by eccentricity.

Current chopping control is utilized to get a steady operation of SRM. According to the configuration of the adopted SRM model, $\theta_{on}$ and $\theta_{off}$ are selected as 28° and 60°, respectively [31]. Based on Equations (9) and (10), it can be deduced that the single-phase UEMFs of SRM in both vertical and longitudinal directions are characterized by different $\varepsilon_x$ and $\varepsilon_z$ values for $\theta_{on} = 28°$ and $\theta_{off} = 60°$. The results are shown in Figure 3. It can be seen that the eccentricity can directly affect UEMF, which is represented in the form of the electromagnetic coupling. To ensure an efficient and stable braking process, the UEMF should be explored as the main internal excitation.

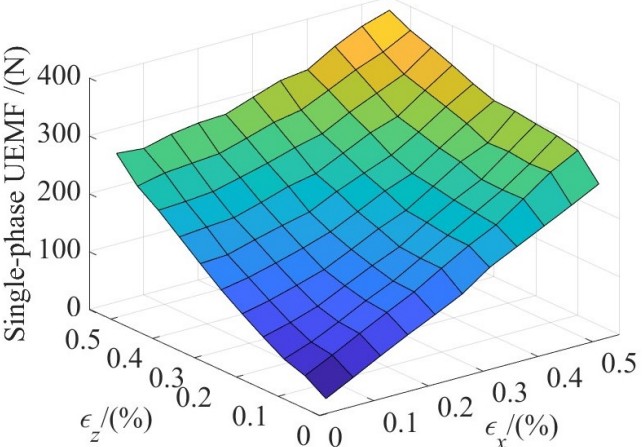

**Figure 3.** The influence of eccentricities on UEMF.

2.1.2. Road Inputs Model

The road inputs model contains road grade (*RG*) and road type (*RT*). *RG* acts on the vehicle as vertical road profiles, while *RT* characterizes the longitudinal road friction.

(1) Vertical road input

The statistical characteristics of *RG* are commonly described by the power spectral density in the vertical direction. The harmonic superposition algorithm is used to generate time-domain road profiles as [46,47]

$$q(t) = \sum_{K=1}^{M} \sqrt{2 \cdot G_q(f_{\text{mid}-K}) \cdot \frac{f_2 - f_1}{M}} \sin(2\pi f_{\text{mid}-K} t + \phi_K) \tag{11}$$

where $q(t)$ is the generated road profile, $f_{\text{mid}-K}$ is the $K$-th middle frequency, $G_q(f_{\text{mid}-K})$ is the power spectral density at $f_{\text{mid}-K}$, $\Phi_K$ is an identifiably distributed phase with the range of $(0, 2\pi)$, and the upper and lower time-domain frequency boundaries are denoted as $f_1$ and $f_2$.

As vehicle speed is variable during braking, the independent variable needs to be changed from time to braking distance by

$$q(t) = q(dis)\big|_{dis=v_c \cdot t} \tag{12}$$

where $v_c$ is the conversion speed and $dis$ is the braking distance at time $t$.

(2) Longitudinal road input

One major parameter used for the longitudinal road input is the road–tire adhesion coefficient-$\mu(\lambda)$, which can be expressed as

$$\mu(\lambda) = c_1\left(1 - e^{-c_2\lambda}\right) - c_3\lambda \tag{13}$$

where $c_1$, $c_2$, and $c_3$ define the road friction conditions (see Table 1) [48] and $\lambda$ is the tire slip ratio. The optimal slip ratio $\lambda^*$ for different road conditions is located in the far right-hand column of Table 1.

**Table 1.** Parameters of different road surfaces.

| Road Type | $c_1$ | $c_2$ | $c_3$ | $\lambda^*$ |
|---|---|---|---|---|
| Dry asphalt | 1.28 | 23.99 | 0.52 | 0.17 |
| Dry cement | 1.20 | 25.17 | 0.54 | 0.16 |
| Wet asphalt | 0.86 | 33.82 | 0.35 | 0.13 |
| Cobblestone | 0.40 | 33.71 | 0.12 | 0.14 |
| Snow | 1.20 | 94.13 | 0.06 | 0.06 |
| Ice | 0.05 | 306.39 | 0 | 0.03 |

*2.2. RTRF Model*

The IWM-tire system consists of a rigid rotor and a deformable tire belt that are connected by springs and dampers in the radial and torsional directions, respectively [49]. By separately capturing the transient road–tire contact patch and tire belt deformation, the RTRF model can accurately reveal the road–tire and tire–rotor forces [50,51]. The RTRF model can be divided into two main parts, i.e., the contact force in the contact patch and the deformation force between the tire belt and the rotor.

2.2.1. Road–Tire Force

(1) Vertical contact force

When a tire is loaded on the road, a large deformation may occur near the contact patch, and a limited contact length that can be simplified to a contact point [50]. Vertical residual stiffness is introduced to obtain the overall vertical tire stiffness [52].

The vertical force at the contact point $F_{cZ}$ is directly related to the vertical residual stiffness $k_{rs}$. By neglecting higher-order terms, a third-order polynomial is used to describe the vertical force due to the residual tire deflection [37], which is given by

$$F_{cZ} = q_{Fzr3} k_{rs}^3 + q_{Fzr2} k_{rs}^2 + q_{Fzr1} k_{rs} \tag{14}$$

where $\omega_t$ is the rotational speed of tire and $q_{Fzr*}$ represents the polynomial coefficients, which can be given by

$$
\begin{cases}
q_{Fzr1} = k_{trd} \frac{q_{Fz1}(1+q_{V2}|\omega_t|)}{k_{trd}-q_{Fz1}(1+q_{V2}|\omega_t|)} \\
q_{Fzr2} = k_{trd} \frac{k_{trd}(k_{trd}\cdot q_{Fz2}+q_{Fzr1}\cdot q_{Fz2})(1+q_{V2}|\omega_t|)}{(k_{trd}-q_{Fz1}(1+q_{V2}|\omega_t|))^2} \\
q_{Fzr3} = 2k_{trd} \frac{q_{Fzr2}\cdot q_{Fz2}(1+q_{V2}|\omega_t|)}{(k_{trd}-q_{Fz1}(1+q_{V2}|\omega_t|))^2}
\end{cases}
\tag{15}
$$

where $q_{V1}$ and $q_{V2}$ are the vertical stiffness correlation coefficients of the tire and $k_{trd}$ is the tire sidewall stiffness.

Then, the residual stiffness is derived as

$$k_{rs} = q(dis_{cp}) - z_t + q_{V1}{\omega_t}^2 \tag{16}$$

where $dis_{cp}$, $z_t$, and $q(dis_{cp})$ are the braking distance, vertical displacement, and effective road roughness at the contact point, respectively.

(2) Longitudinal contact force

For the longitudinal tire model, as the deformation of the sidewall is already represented by the displacement of the tire belt, only a slip model for the contact patch is required. The elastic deformation of the tire sidewall results in a difference between the rotor's and the tire's linear velocity in the contact patch [53,54]. The delay of the contact patch's response to a change in slip (the relaxation length of the contact patch) is approximated with a first-order filter, which is given by [55]

$$\sigma_c \dot{\lambda}_d + |v_{cr}|\lambda_d = -v_{c,sx} \tag{17}$$

where $\lambda_d$ is the tire slip ratio calculated by the first-order filter, $\sigma_c$ is the relaxation length of the contact patch, and $v_{cr}$ is the linear velocity at the contact patch. The parameter $v_{c,sx}$ is the slip velocity at the contact patch, while $v_{cr}$ and $v_{c,sx}$ are expressed as

$$v_{cr} = r_e \omega_t, \quad v_{c,sx} = v_t - r_e \omega_t \tag{18}$$

where $r_e$ is the effective rolling radius and $v_t$ is the forward velocity of the wheel center. The slip velocity of an elastic tire can be defined as the absolute speed of an imaginary point. The effective rolling radius $r_e$ is defined such that the slip velocity is zero for free rolling. The effective rolling radius can be expressed in the form of a third-order polynomial, which is given by

$$r_e = q_{re3}\sqrt{F_{cZ}^3} + q_{re2}\sqrt{F_{cZ}^2} + q_{re1}\sqrt{F_{cZ}} + q_{re0} + q_{V1}{\omega_t}^2 \tag{19}$$

where $q_{re0}$, $q_{re1}$, $q_{re2}$, and $q_{re3}$ are the rolling correlation coefficients.

Under full adhesion, $\sigma_c$ is equal to half of the contact length [56], which is given by

$$
\begin{cases}
F_Z = (m_s + m_{ms} + m_{mr} + m_t)g - F_{cZ} \\
\sigma_c = \frac{q_{a2}\sqrt{F_Z^2}+q_{a1}\sqrt{F_Z}}{2}
\end{cases}
\tag{20}
$$

where $F_Z$ is the normal force, $m_s$ is the quarter sprung mass of the vehicle, $m_{mr}$ is the mass of the motor, $m_{ms}$ is the stator and axle mass, $m_t$ is the mass of the tire, and $q_{a1}$ and $q_{a2}$ represent the contact length correlation coefficients. According to Equations (19) and (20), the vertical forces $F_{cZ}$ and $F_Z$ play decisive roles in determining the vertical–longitudinal road–tire coupling effect.

A friction model is used to describe the longitudinal forces at the contact patch as a function of the slip ratio. The empirical magic formula is adopted, which is provided by [36]

$$F_{cX} = F_Z \mu(\lambda_d) \sin(C \arctan(B\lambda_d - E(B\lambda_d - \arctan(B\lambda_d)))) \tag{21}$$

where $F_{cX}$ is the longitudinal friction force, $B$, $C$, and $E$ are the model parameters. It can be observed that the slip ratio has an enormous influence on the longitudinal tire force and that the longitudinal and vertical motions are highly coupled.

In practice, the slip ratio is defined according to the difference between the longitudinal velocity of the vehicle and the linear velocity of the wheel, which can be obtained by

$$\lambda_c = \frac{v_{ms} - \omega_{mr} R_t}{v_{ms}} \tag{22}$$

where $\lambda_c$ is the measured slip ratio, $v_{ms}$ is the longitudinal vehicle velocity, and $\omega_{mr}$ is the rotational speed of the rotor. It is worth noting that the changes in the effective rolling radius and tire contact patch are ignored, as these would introduce significant errors into the dynamic process. The relative error of the slip ratio estimation is defined as

$$e_{silp}(t) = \lambda_d(t) - \lambda_c(t) \tag{23}$$

where $e_{slip}$ is the error between the slip ratios from the established and the conventional model, which is used as an important indicator for vehicle dynamics performance in the later analysis.

### 2.2.2. Tire–Rotor Force

Although $q(dis)$ is taken as the vertical input from road profiles, it may induce tire deformation in both the vertical and longitudinal directions. The tire force model can be described by a finite number of independent radial spring and damping elements evenly distributed in the lower semicircle, as shown in Figure 4 [56]. The total number of the discrete radial elements is denoted by $N_{tr}$ [57]. In Figure 4, $z_{mrc}$ is the vertical coordinate at the center of the rotor, $z_t$ is the vertical displacement of the contact point, and $\alpha_i$ represents the angle between an arbitrarily chosen element and the vertical that ranges from $-\arcsin(\sigma_c/R_t)$ to $+\arcsin(\sigma_c/R_t)$. For the positive $x$-axis, the subscript $i$ indicates the sequence number of the element that ranges from 1 to $N_{tr}$, while $\gamma_{\alpha i}$, $dis_{\alpha i}$, and $q(dis_{\alpha i})$ are the radial deformation, braking distance, and road elevation of the element, respectively. As a result of tire deformation, the vertical component $F_{tz\alpha i}$ and longitudinal component $F_{tx\alpha i}$ of the radial spring and damping element forces are provided by

$$\begin{cases} F_{tz\alpha i} = \begin{cases} (\gamma_{\alpha i} k_{trd} + \dot{\gamma}_{\alpha i} c_{trd}) \cos \alpha i & \gamma_{\alpha i} > 0 \\ \dot{\gamma}_{\alpha i} c_{trd} \cos \alpha i & \gamma_{\alpha i} \leq 0 \end{cases} \\ F_{tx\alpha i} = \begin{cases} (\gamma_{\alpha i} k_{trd} + \dot{\gamma}_{\alpha i} c_{trd}) \sin \alpha i & \gamma_{\alpha i} > 0 \\ \dot{\gamma}_{\alpha i} c_{trd} \sin \alpha i & \gamma_{\alpha i} \leq 0 \end{cases} \end{cases} \tag{24}$$

where $c_{trd}$ is the tire sidewall translational damping. The deformation and deformation rate of a certain radial element can be approximately calculated by

$$\begin{cases} \alpha i = -\arcsin(\sigma_c/R_t) + 2i \arcsin(\sigma_c/R_t)/N_{tr} \\ dis_{\alpha i} = x_t + \sqrt{(R_t^2 - \sigma_c^2)} \cdot \tan \alpha i \\ (R_t - \gamma_{\alpha i}) \cos \alpha i + q(dis_{\alpha i}) = R_t + z_{mrc} + z_t \end{cases} \tag{25}$$

where $x_t$ is the longitudinal displacement of the tire. The vertical component $F_{tz}$ and longitudinal component $F_{tx}$ of the tire–rotor force caused by tire deformation at the rotor's center can be obtained by summing the force components of each spring and damping element. These can be obtained by

$$\begin{cases} F_{tz} = \sum_{i=1}^{N_{tr}} F_{tx\alpha i} \\ F_{tx} = \sum_{i=1}^{N_{tr}} F_{tz\alpha i} \end{cases} \tag{26}$$

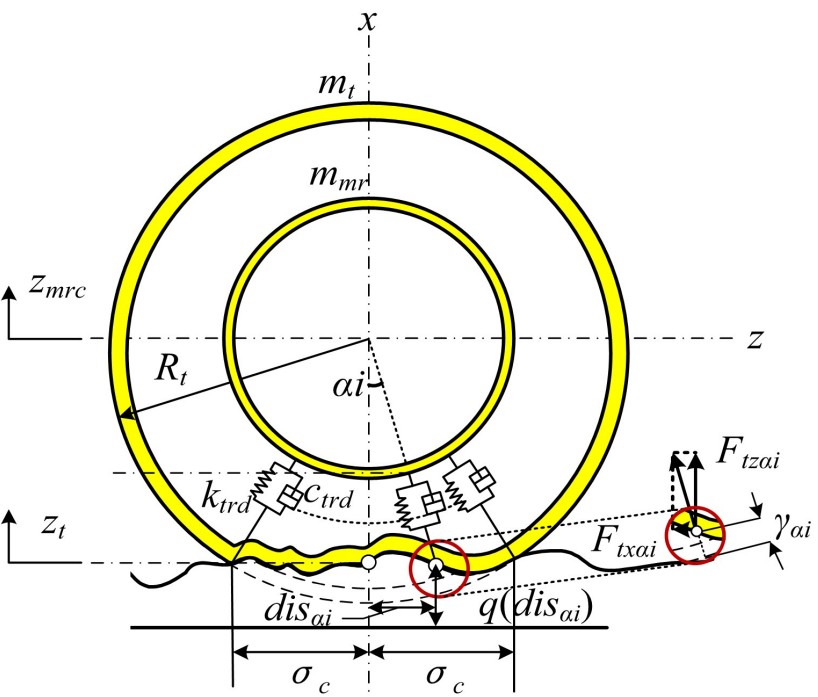

**Figure 4.** The tire–rotor force.

### 2.3. NTDC Model

To reveal the vibration mechanism of IWMD EVs under the electromagnetic coupling, the NTDC model describing the vertical, longitudinal, and rotational motions is established as shown in Figure 5. Based on the Newton's law, the equation of vibration is given by

$$\begin{cases} m_t \ddot{z}_t + F_{tz} + F_{cZ} = 0 \\ m_{mr} \ddot{z}_{mr} - F_{tz} + F_{bv} - F_{uv} = 0 \\ m_{ms} \ddot{z}_{ms} + F_{sus} - F_{bv} + F_{uv} = 0 \\ m_s \ddot{z}_s - F_{sus} = 0 \\ F_{bv} = k_{beaz}(z_{mr} - z_{ms}) \\ F_{sus} = c_{sus}(\dot{z}_{ms} - \dot{z}_s) + k_{trd}(z_{ms} - z_s) \\ q_{a*} m_t \ddot{x}_t - F_{cX} + F_{tl} + F_{roll} = 0 \\ m_{mr} \ddot{x}_{mr} - F_{tl} - F_{bl} + F_{ul} = 0 \\ m_{ms} \ddot{x}_{ms} + F_{bl} - F_{ul} + F_{usl} = 0 \\ m_s \ddot{x}_s - F_{usl} + F_{air} = 0 \\ F_{tl} = c_{trd}(\dot{x}_t - \dot{x}_{mr}) + k_{trd}(x_t - x_{mr}) - F_{tx} \\ F_{bl} = k_{bear}(x_{mr} - x_{ms}) \\ F_{usl} = c_{ux}(\dot{x}_{ms} - \dot{x}_s) + k_{ux}(x_{ms} - x_s) \\ F_{roll} = \mu_{roll} \dot{x}_t \\ F_{air} = k_{air} \dot{x}_s^2 \end{cases} \quad \begin{cases} I_t \ddot{\theta}_t + T_t + r_e F_{cX} = 0 \\ I_{mr} \ddot{\theta}_{mr} - T_t - T_b - T_{IWM} = 0 \\ T_t = c_{trt}(\dot{\theta}_t - \dot{\theta}_{mr}) + k_{trt}(\theta_t - \theta_{mr}) \end{cases} \tag{27}$$

where $I_t$ is the tire ring inertia, $I_m$ is the rim and rotor inertia, $k_{trt}$ is the tire sidewall rotational stiffness, $c_{trt}$ is the tire sidewall rotational damping, $k_{bear}$ is the motor bearing stiffness, $k_{sus}$ is the suspension stiffness, $c_{sus}$ is the suspension damping, $k_{ux}$ is the shaft sleeve stiffness, $c_{ux}$ is the shaft sleeve damping, $z_i$ is the vertical displacement of components, $F_{bv}$ and $F_{bl}$ are the vertical and longitudinal motor bearing forces, $F_{sus}$ is the suspension vertical force, $x_i$ is the longitudinal displacement of components, $F_{tl}$ is the

longitudinal tire force, $F_{usl}$ is the longitudinal suspension and shaft sleeve force, $F_{air}$ and $F_{roll}$ are the rolling resistance and air drag, $\mu_{roll}$ and $k_{air}$ are the coefficients of $F_{air}$ and $F_{roll}$, $\theta_i$ is the rotational angle, $T_t$ is the internal tire torque, $T_{IWM}$ is the braking torque of the motor, and $T_b$ is the braking torque of the brake caliper.

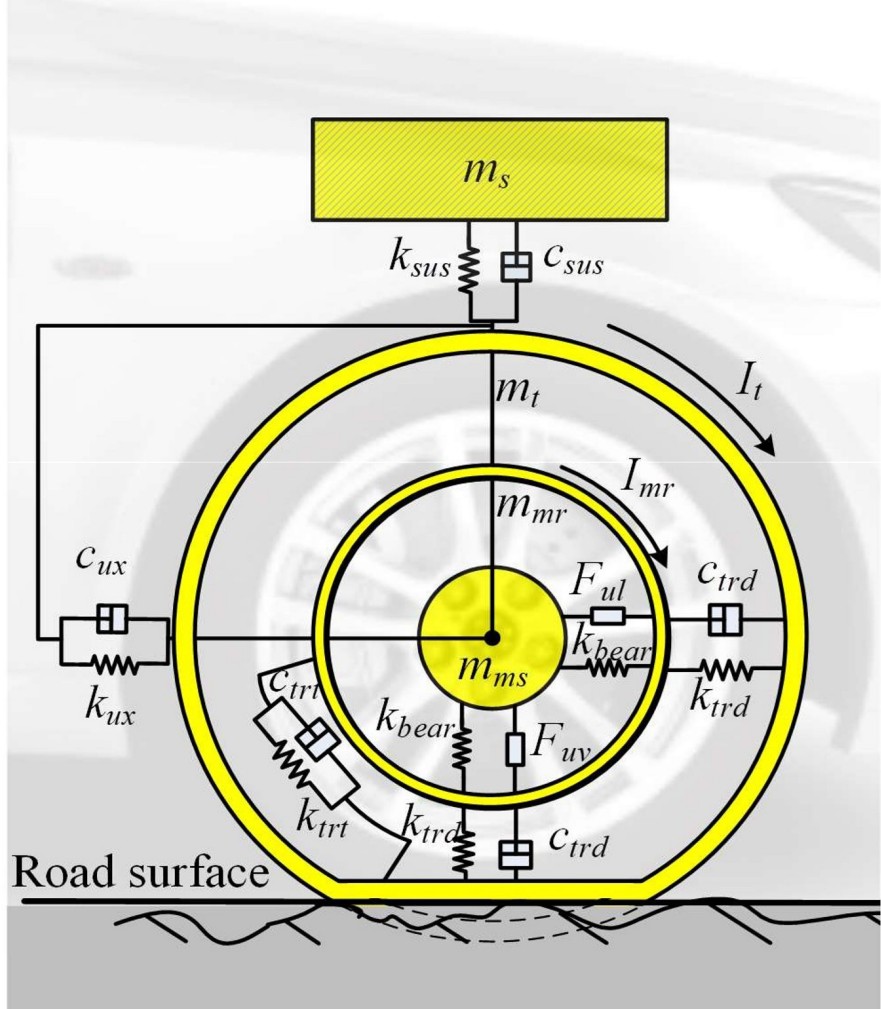

**Figure 5.** The NTDC vibration coupling model.

The schematic of the vibration model is illustrated in Figure 6. Theoretically, the vertical and longitudinal coupling effect occurs in UEMF during the braking process. As previously mentioned, $\varepsilon_z$ in the vertical direction and $\varepsilon_x$ in the longitudinal direction represent the eccentricity of motor due to the road surface roughness and braking torque, respectively, while $\varepsilon_z$ and $\varepsilon_x$ synthesize the radial eccentricities to produce the radial UEMF by decoupling phase structure angle; that is to say, the coupling effect between $\varepsilon_z$ and the IWM system produces $F_{uv}$. The vertical UEMF and eccentricity are characterized by the coupling effect under road inputs. Similar to the vertical process, $F_{cX}$ directly influences $\varepsilon_x$ in the longitudinal vibration dynamics, which further produces $F_{ul}$ within its coupling process in the IWM system.

The vertical–longitudinal coupling effect exists in the RTRF model as well. External excitations have two main sources, i.e., *RG* and *RT*. For a given road profile, the longitudinal vehicle velocity determines the vertical road input $z_r$ (*RG*). In the road–tire force model, the vertical load of the tire is a direct determinant of $r_e$ and $\sigma_c$. For the tire–rotor force model, the radial deformation forces are decoupled into vertical and longitudinal forces, which serve as the inputs to the vertical and longitudinal vibration dynamics modules. In the

SIWMS model, the vertical–longitudinal coupling effect is significant due to the unique structure of IWM.

**Figure 6.** The vertical–longitudinal coupling in the quarter-SIWMS model with the red fonts indicating the coupling variables.

## 3. Simulation and Verification of the Quarter-SIWMS Model

### 3.1. SIWMS Simulation Model

According to the established SIWMS model, a simulation platform was developed in the MatLab /Simulink. Braking maneuvers were performed for model verification. The specifications of the test vehicle are listed in Table 2. The tire is a summer tire for passenger cars, which is designated as 205/55R16 [58].

**Table 2.** The specifications of the test vehicle.

| Parameters | Value | Parameters | Value | Parameters | Value |
|---|---|---|---|---|---|
| Effective rolling radius | | Vertical sidewall stiffness | | IWM and vehicle parameters | |
| $R_t$ | 0.3160 m | $q_{V1}$ | $8.5352 \times 10^{-8}$ m s$^2$ | $c_{ux}$ | 1800.39 N·s/m |
| $q_{re0}$ | 0.3164 m | $q_{V2}$ | $8.81 \times 10^4$ s | $k_{ux}$ | $2.5 \times 10^4$ N/m |
| $q_{re1}$ | $-9.3972 \times 10^{-4}$ m | $q_{Fz1}$ | $1.4389 \times 10^5$ N/m | $k_{sus}$ | $3.2 \times 10^4$ N/m |
| $q_{re2}$ | $-9.3972 \times 10^{-6}$ m | $q_{Fz2}$ | $4.5090 \times 10^6$ N/m$^2$ | $c_{sus}$ | $1.8 \times 10^3$ N·s/m |
| $q_{re3}$ | $-3.0346 \times 10^{-8}$ m | | | $k_{bear}$ | $2.08 \times 10^7$ N/m |
| Contact patch | | Tire parameters | | $k_{air}$ | $5.3 \times 10^{-3}$ N/m |
| $q_{a1}$ | $1.478 \times 10^{-3}$ m | $I_t$ | 0.546 kg·m$^2$ | $m_{ms}$ | 9.5 kg |
| $q_{a2}$ | $-5.6829 \times 10^{-6}$ m/N | $I_{mr}$ | 0.417 kg·m$^2$ | $m_{mr}$ | 22.5 kg |
| $\mu_{roll}$ | 0.015 N·s/m | $m_t$ | 6.15 kg | $m_s$ | 332 kg |
| Magic Formula | | $c_{trd}$ | 510 N·s/m | | |
| $B$ | 20.2937 | $k_{trd}$ | $1.8 \times 10^6$ N/m | | |
| $C$ | 1.9655 | $k_{trt}$ | $5.1 \times 10^4$ N/m | | |
| $E$ | 0.8613 | $c_{trt}$ | $20 \times$ N·s/m | | |

For road inputs, four typical conditions were taken as the excitation sources acting on the SIWMS model. They are the ISO-A, ISO-B, dry asphalt, and cobblestone as shown in Figure 7. The road roughness was transformed from time-dependent to braking distance-dependent, as shown in Figure 7a. Sequentially, the modification of each response variable was then obtained under the excitation sources.

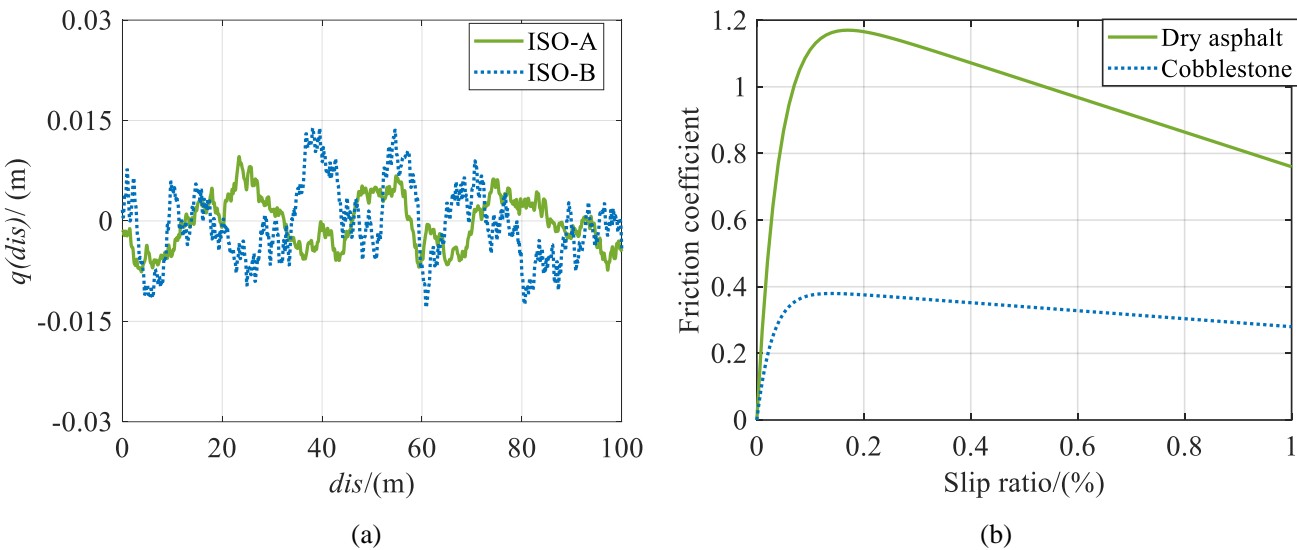

**Figure 7.** Four typical conditions of road inputs: (**a**) *RG* and (**b**) *RT*.

### 3.2. Brake Control System

The braking torque is provided by the hydraulic braking system (HBS). The vehicle control unit (VCU) obtains the rotational speed of IWM and calculates the slip ratio $\lambda_c$ according to Equation (22). Then, the VCU delivers the difference between the optimal and the actual slip ratio to the ABS module at each time step.

The initial longitudinal vehicle velocity was set to be 60 km/h and the ABS cut-off vehicle velocity was set to be 5 km/h. Before verifying the proposed model, the efficacy of the PID controller for optimal slip ratio tracking needs be verified. The control performance for a vehicle traveling on the ISO-A, ISO-B, dry asphalt, and cobblestone road conditions while executing braking at 60 km/h is shown in Figure 8. It can be seen that the PID controller is able to closely track the desired slip ratios under different road frictions.

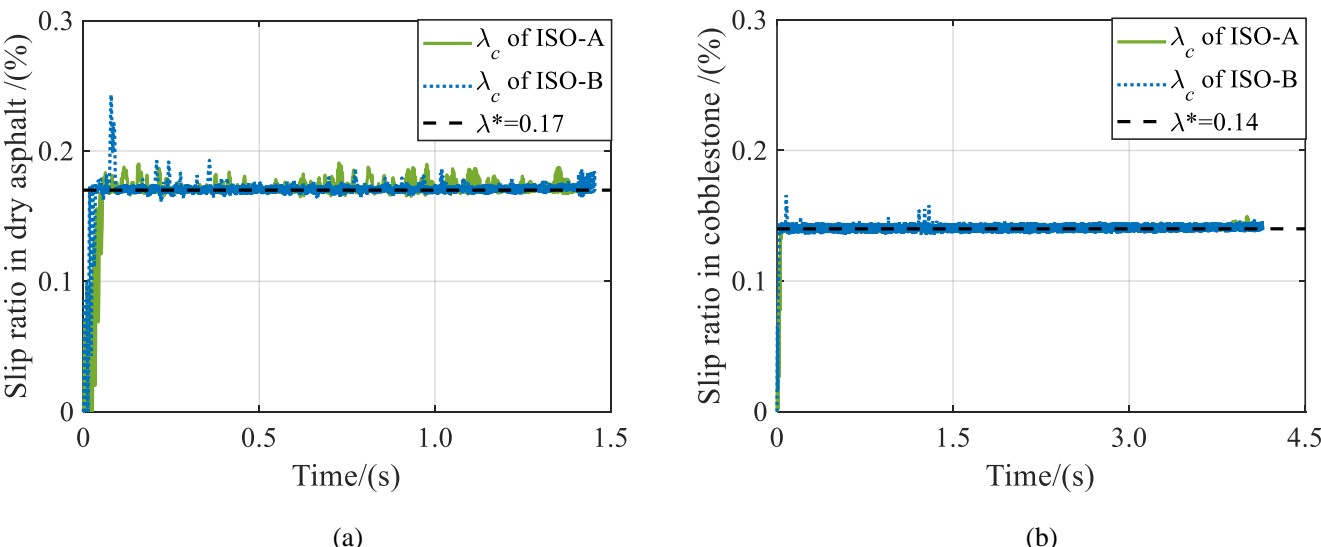

**Figure 8.** The actual slip ratio on (**a**) the dry and (**b**) the cobblestone road during optimal ratio tracking control.

### 3.3. Vertical–Longitudinal Dynamics Evaluation Indexes

To examine the vertical dynamics during the braking process, the root mean squared (RMS) errors of the vertical acceleration (RVA) and of the tire dynamic load (RTDL) were

selected to represent the vehicle response, which can reflect the ride comfort and handling performance, respectively. The longitudinal vibration is an important indicator of driving performance assessment. Usually, the longitudinal vibration is characterized by the frequency and attitude or power spectrum of the longitudinal vehicle acceleration or jerk (the derivative of acceleration), which happens with greater amplitude in specific transient operating conditions such as rapid acceleration and emergency braking [59]. Therefore, the longitudinal acceleration fluctuation rate (LAFR) was selected, which is provided by

$$\text{LAFR} = \frac{\sqrt{\frac{1}{t}\int_0^t \left[a_x(t) - \frac{1}{t}\int_0^t (a_x(t)\mathrm{d}t)^2\right]\mathrm{d}t}}{\frac{1}{t}\int_0^t (a_x(t)\mathrm{d}t)\mathrm{d}t} \tag{28}$$

where $a_x$ is the instantaneous longitudinal vehicle acceleration and $t$ is the braking time.

The tire slip ratio is a key parameter during braking. For the IWM system, the tire slip ratio fluctuates more dramatically due to the torque fluctuation and UEMF. In addition, the increase of inertia around the axle can aggravate the deviation from the true value. To precisely measure the error, the signal-to-noise ratio (SNR) of the slip ratio is introduced for evaluation, which is given by

$$\text{SNR} = \frac{\sqrt{\frac{1}{t}\int_0^t e_{slip}{}^2(t)\mathrm{d}t}}{\frac{1}{t}\int_0^t \lambda_c(t)\mathrm{d}t} \tag{29}$$

*3.4. Virtual Prototype Validation for the SIWMS*

In this study, a virtual prototype was constructed to establish a high-fidelity multi-body model using the CATIA, ADAMS, and Matlab/Simulink environment.

The developed virtual prototype is shown in Figure 9 for demonstration. First, a complete vehicle model was developed in CATIA based on the vehicle specifications obtained from an actual IWMD EV. Then, the SIWMS model was integrated into ADAMS and the constraints of each component were established. The kinematic joints of each component were defined and the loads and drives were added. Finally, the braking torques of the caliper and IWM in ADAMS were taken from the brake controller and IWM system modules in the MatLab/Simulink, respectively. The radial UEMF was applied on the rotor and stator surfaces of IWM. The vehicle responses of the virtual prototype model and the numerical model of the SIWMS in the vertical and longitudinal directions are compared in Figure 10. The error statistics are listed in Table 3.

**Table 3.** The selected evaluation indexes.

| Response Variables | Numerical Model | VP Model | Error |
|---|---|---|---|
| RVA/(m/s$^2$) | 0.4353 | 0.4157 | 4.71% |
| RTDL/(N) | 296.5 | 282.6 | 4.92% |
| LAFR/(m/s$^2$) | 10.77 | 11.03 | 2.36% |
| SNR/(%) | 31.17 | 32.55 | 4.24% |

As shown in Figure 10a–d, the two models demonstrate the same trend in all response variables with marginal lag time. The main distinction between the two models lies in the longitudinal direction. In particular, the proposed SIWMS model is validated by comparing the selected evaluation indexes as shown in Table 3. It can be seen that all the errors are within 5%, which verifies the validity of the proposed model [60].

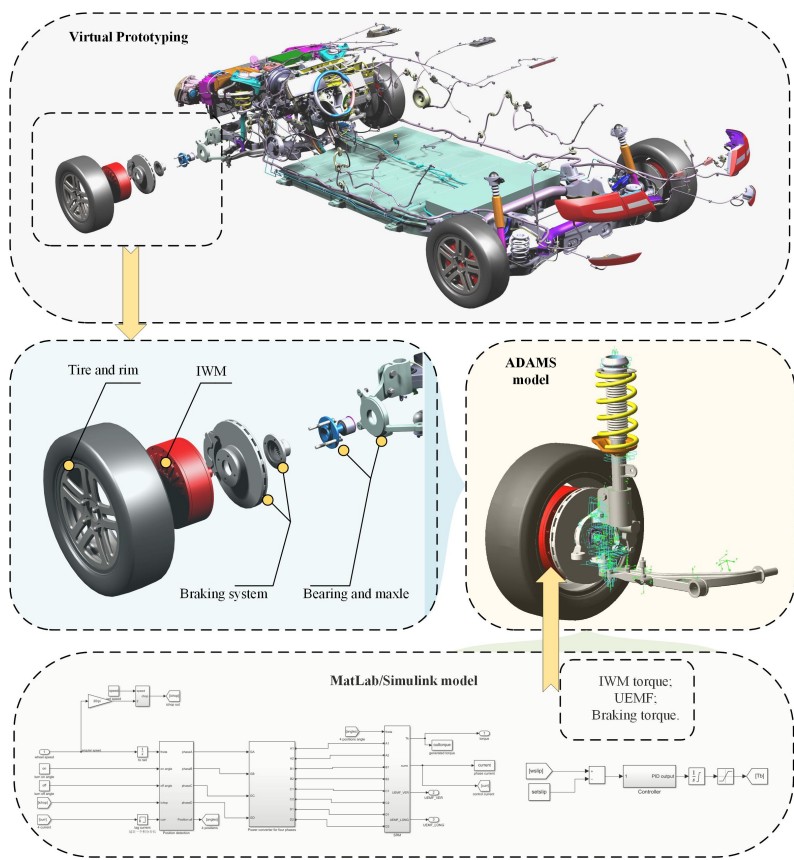

**Figure 9.** The build process of the VP model.

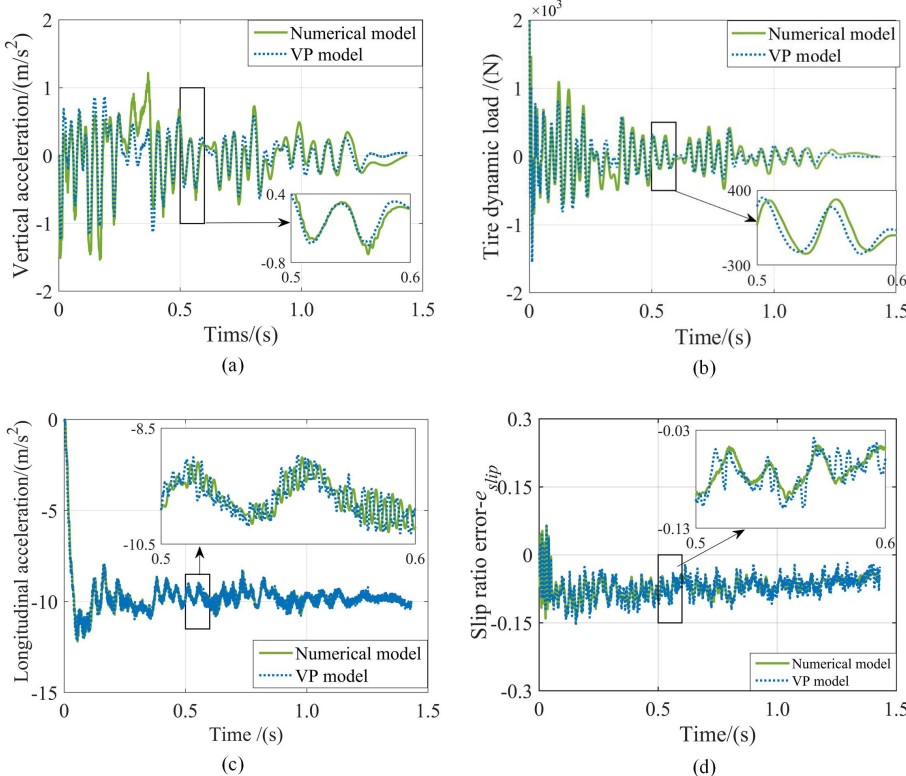

**Figure 10.** Simulation results obtained from the virtual prototype model and the proposed numerical model of SIWMS: (**a**) vertical acceleration of the vehicle; (**b**) dynamic load of the tire; (**c**) slip ratio on the dry asphalt road; (**d**) slip ratio error-$e_{slip}$.

## 4. UEMF Effects and SIWMS Optimization

In this section, a simulation comparison is performed with and without considering UEMF. Four key parameters are selected and their effects on vehicle vertical–longitudinal dynamics are investigated. Furthermore, the key stiffness and damping parameters are optimized using a multi-objective optimization algorithm.

### 4.1. Analysis of UEMF Effects

4.1.1. Relationship between UEMF and Eccentricity

Due to the resultant UEMF, the dynamic eccentricity of the IWM due to the road input and slip ratio fluctuations has a remarkable impact on vehicle dynamics. To obtain the vertical and longitudinal eccentricity values of the IWM during braking and the vertical and longitudinal UEMFs generated by the IWM, numerical analysis was performed using the model validated in the previous subsection. The results are shown in Figure 11.

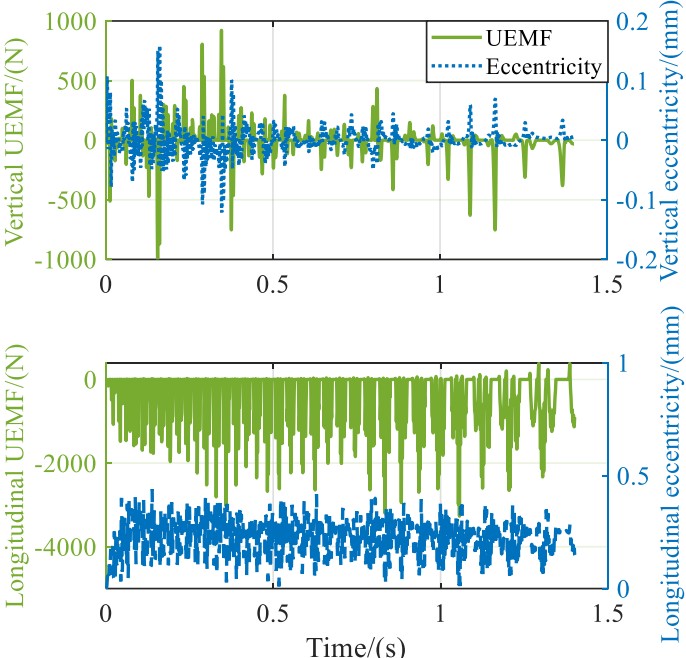

**Figure 11.** The UEMF and eccentricity calculation results.

It can be seen that the IWM has dynamic eccentricities and UEMFs in both the vertical and longitudinal directions. In the vertical direction, the eccentricity is random. This leads to a random state of the vertical UEMF that varies from $-1000$ N to 1000 N. In the longitudinal direction, the eccentricity between the stator and the rotor is always positive, as the overall acceleration is negative during the braking process. Moreover, its corresponding longitudinal UEMF is negative and varies from 0 N to 2500 N, with the same order of magnitude as in the vertical direction.

4.1.2. UEMF Effects

The UEMF has a significant effect on vehicle dynamics. Additional simulation results are shown in Table 4.

**Table 4.** Evaluation indexes for vehicle dynamics during braking.

| | With UEFM | Without UEMF | Percentage of Degradation | | With UEFM | Without UEMF | Percentage of Degradation |
|---|---|---|---|---|---|---|---|
| ISO-A and Dry asphalt | | | | ISO-B and Dry asphalt | | | |
| RVA (m/s$^2$) | 0.4353 | 0.4325 | 0.65 | RVA (m/s$^2$) | 0.9081 | 0.8935 | 1.63 |
| RTDL (N) | 296.5 | 295.3 | 0.41 | RTDL (N) | 631.2 | 622.2 | 1.45 |
| LAFR (%) | 10.77 | 10.66 | 1.03 | LAFR (%) | 12.82 | 12.27 | 4.48 |
| SNR (%) | 31.17 | 30.67 | 1.63 | SNR (%) | 36.58 | 34.61 | 5.69 |
| ISO-A and Cobblestone | | | | ISO-B and Cobblestone | | | |
| RVA (m/s$^2$) | 0.4764 | 0.4734 | 0.63 | RVA (m/s$^2$) | 0.9413 | 0.931 | 1.11 |
| RTDL (N) | 321.8 | 320.5 | 0.41 | RTDL (N) | 700.3 | 687.3 | 1.89 |
| LAFR (%) | 6.937 | 6.796 | 2.07 | LAFR (%) | 10.26 | 9.72 | 5.56 |
| SNR (%) | 31.49 | 30.76 | 2.37 | SNR (%) | 37.22 | 34.79 | 6.98 |

According to the comparison results shown in Table 4, the RVA and RTDL are slightly increased when the effect of UMF is considered, while the increases in LAFR and SNR are more profound. This shows that the UEMF caused by magnet gap deformation negatively affects the ride comfort and longitudinal stability of the vehicle. During braking, UEMF has a greater effect on the longitudinal dynamics than on the vertical dynamics. Moreover, the effects on the ISO-B and cobblestone roads are more noticeable.

### *4.2. Sensitivity Analysis and Optimization*

In this section, the influence of the IWM system parameters on vertical–longitudinal dynamics is investigated and a multi-objective optimization approach is employed to optimize the system parameters.

#### 4.2.1. Sensitivity Analysis

To meet multiple design requirements, a comprehensive sensitivity analysis method is applied to evaluating the influence of each vehicle parameter on the selected optimized objectives including RVA, RTDL, LAFR, and SNR. According to previous investigations [31,61] and our preliminary analysis, $k_{sus}$, $c_{sus}$, $k_{trt}$, and $k_{bear}$ were selected as the key parameters to be optimized. The original parameter values listed in Table 2 were taken as the benchmark and each value was set to 0.5/1/1.5/2 times the benchmark value, respectively. The numerical analysis was conducted while keeping the other parameters constant. To ensure that the response quantities of each index were comparable, normalized dimensionless processing was performed using

$$x^*(i) = \frac{x(i) - \eta}{\sigma} \tag{30}$$

where $x^*(i)$ is the converted value, $x(i)$ is the initial value, and $\eta$ and $\sigma$ are the mean and standard deviations of all sample data. Generally, a larger value of $x^*(i)$ indicates more significant influence on the optimization objectives, while the positive curve slope indicates that the optimization objective will increase with the increasing design variable and vice versa. Thus, the sensitivity of each selected design parameter on the optimization objectives can be calculated. The results are shown in Figure 12.

As seen from Figure12, each parameter exhibits different levels of sensitivity to the optimization objectives. Specifically, the RVA and LAFR demonstrate similar increasing trends, with $k_{sus}$ increasing from 0.5 up to twice the initial value. For $c_{sus}$ and $k_{bear}$, the RTDL and SNR reflect a constant trend of sensitivity, with $c_{sus}$ being negative and $k_{bear}$ being positive. However, the evaluation indexes with the same trend for $k_{trt}$ are LAFR and SNR.

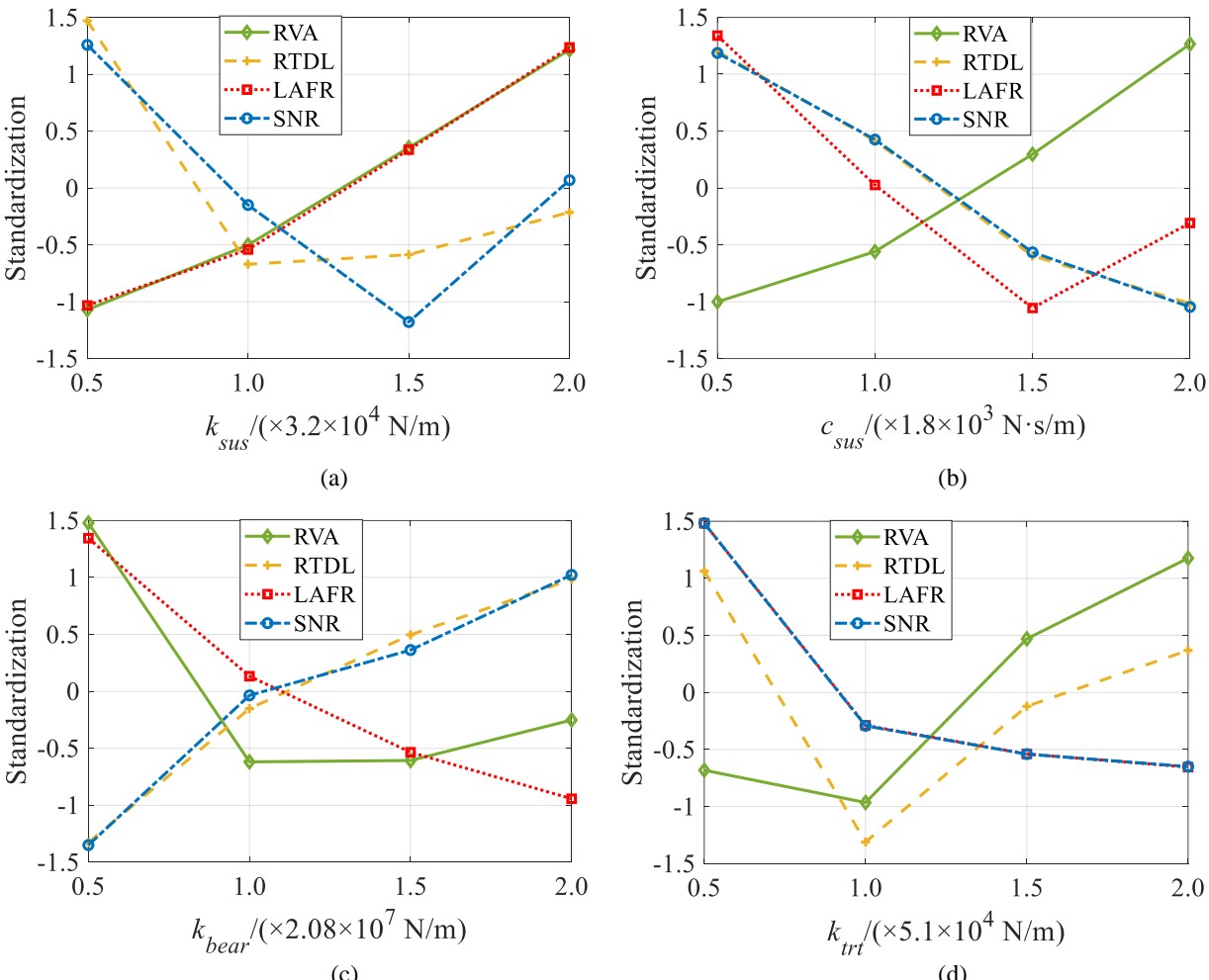

**Figure 12.** The influence of the IWM system parameters on dynamics evaluation indexes: (**a**) $k_{sus}$; (**b**) $c_{sus}$; (**c**) $k_{bear}$; (**d**) $k_{trt}$.

It can be observed that key parameters have a strong sensitivity to the vertical–longitudinal dynamics of IWMD EVs. Moreover, typical vertical parameters influence the longitudinal vehicle dynamics, while longitudinal parameters affect the vertical vehicle dynamics. However, the impact is not consistent, and is sometimes even contradictory. It is relatively difficult to determine appropriate parameter values based on the sensitivity results. To improve the results, the characteristics used for these dynamics should be selected while taking the target requirements into consideration.

### 4.2.2. Multi-Objective Optimization of Key Parameters

To effectively solve the problem mentioned in the previous section, a technique for order preference by similarity to ideal solution (TOPSIS) is adopted to quantitatively investigate the sensitivity of each design parameter [62]. First, the decision matrix is defined as

$$D = \begin{bmatrix} x_{11} & x_{12} & .. & x_{1n} \\ x_{21} & x_{22} & .. & x_{2n} \\ \vdots & \vdots & \vdots & \vdots \\ x_{m1} & x_{m2} & .. & x_{mn} \end{bmatrix} \tag{31}$$

where $x_{ij}$ is the value for each criterion. Next, the decision matrix is normalized using

$$r_{ij} = \frac{x_{ij}}{\sqrt{\sum\limits_{i=1}^{m} x_{ij}^2}} \tag{32}$$

In order to establish a normal compatible matrix, the weights of each criterion are multiplied in a normalized matrix provided by

$$v_{ij} = r_{ij} \times w_j \tag{33}$$

where $w_j$ is the weighting factor symmetric to the $j$th criterion. The weighting factor can be expressed as

$$\sum_{j=1}^{n} w_j = 1 \tag{34}$$

The positive and negative desired values are derived as

$$
\begin{aligned}
A^+ &= \left\{ \left( \max_i v_{ij} \mid j \in \Omega_b \right), \left( \min_i v_{ij} \mid j \in \Omega_c \right) \right\} \\
&= \left\{ v_j^+ \mid j = 1, 2, \ldots, n \right\} \\
A^- &= \left\{ \left( \min_i v_{ij} \mid j \in \Omega_b \right), \left( \max_i v_{ij} \mid j \in \Omega_c \right) \right\} \\
&= \left\{ v_j^- \mid j = 1, 2, \ldots, n \right\}
\end{aligned}
\tag{35}
$$

where $\Omega_b$ is related to the positive indicators and $\Omega_c$ is related to the negative ones.

The Euclidean distances from the positive and negative desired values are calculated using

$$
\begin{cases}
d_i^+ = \sqrt{\sum\limits_{j=1}^{n} \left( v_{ij} - v_j^+ \right)^2} \\
d_i^- = \sqrt{\sum\limits_{j=1}^{n} \left( v_{ij} - v_j^- \right)^2}
\end{cases}
\tag{36}
$$

Here, the relative proximity of each option, defined in terms of the closeness value $Cl$, is considered as the desired solution, which is provided by

$$Cl_i^* = \frac{d_i^-}{d_i^- + d_i^+} \tag{37}$$

During the braking process, longitudinal vehicle dynamics control is considered. The effects of UEMF on RVA and RTDL are not apparent. The weighting coefficient $w_1 = 0.15$ is taken for RVA while $w_2 = 0.15$ for RTDL, $w_3 = 0.3$ for LAFR and $w_4 = 0.4$ for SNR.

Each initial value is provided in Table 2, and the variation range is 0.5–2 times the initial value. The optimal values obtained by the optimization are $k_{sus} = 2.61 \times 10^4$ N/m, $c_{sus} = 2.65 \times 10^3$ N·s/m, $k_{bear} = 2.87 \times 10^7$ N/m, and $k_{trt} = 7.32 \times 10^4$ N/m. The optimization results of vehicle dynamics characteristics are shown in Figure 13a–d, while the variations of the evaluation indexes are shown in Table 5.

**Table 5.** Optimization results of evaluation indexes.

|  | RVA (m/s$^2$) | RTDL (N) | LAFR (%) | SNR (%) |
|---|---|---|---|---|
| Before optimization | 0.9423 | 700.3 | 10.26 | 37.22 |
| After optimization | 0.9147 | 674.6 | 9.74 | 34.94 |
| Optimization results | 2.93% ↓ | 3.67% ↓ | 5.07 % ↓ | 6.13% ↓ |

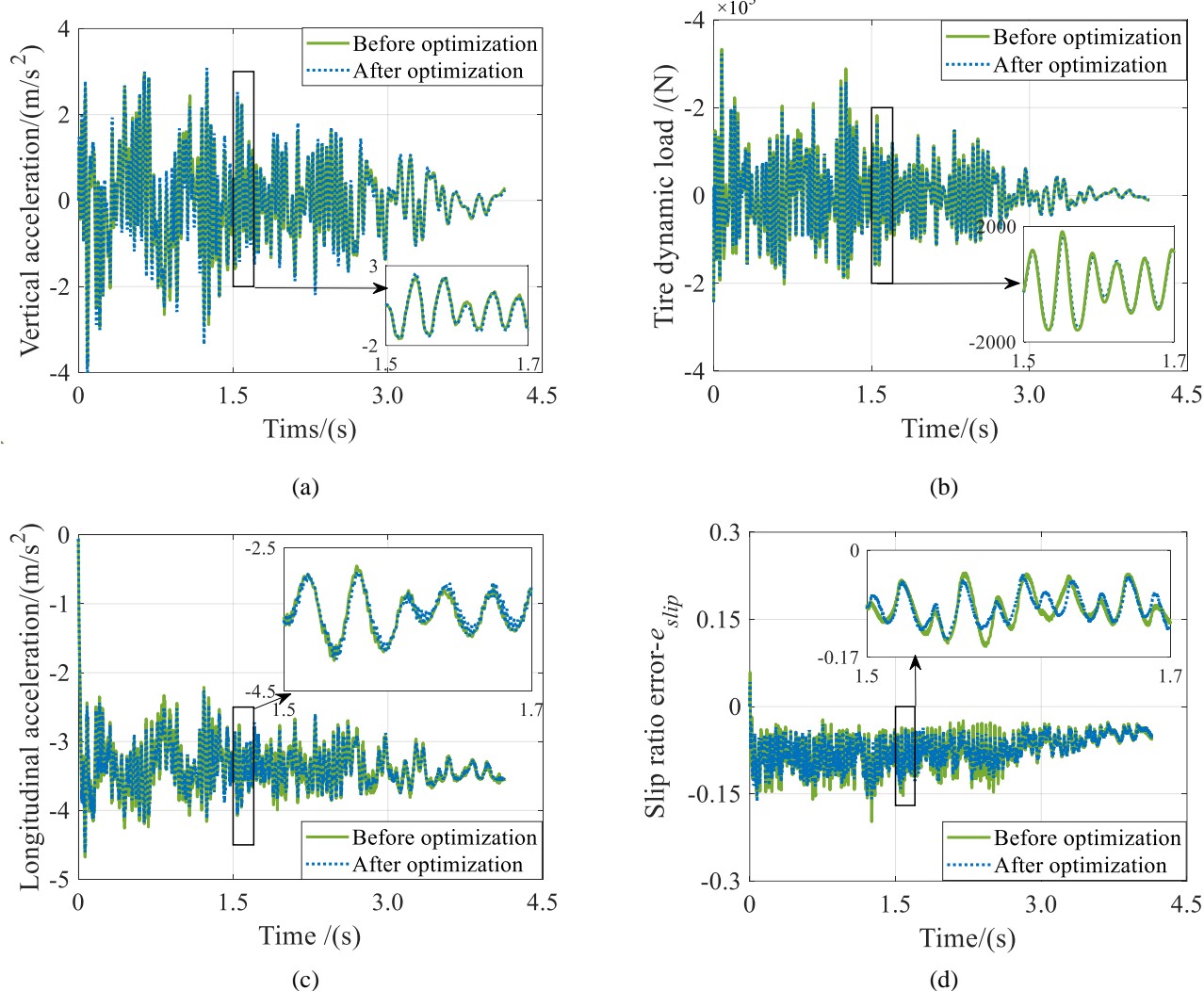

**Figure 13.** Optimization results of vehicle dynamics characteristics: (**a**) vertical acceleration of the vehicle; (**b**) dynamic load of the tire; (**c**) longitudinal acceleration of the vehicle; (**d**) slip ratio error $e_{slip}$.

All four evaluation indexes vary from 2.93% to 6.18% and are reduced after optimization. Particularly, the SNR and LAFR are reduced by 6.13% and 5.07%, respectively. The optimization results indicate that the vertical and longitudinal dynamics are improved after the proposed TOPSIS optimization and that the UEMF effects are compensated for to a certain extent. In addition, according to Figure 13, the fluctuations of vertical acceleration, dynamic load of tire, longitudinal acceleration, and slip ratio error are all suppressed. Moreover, the vertical dynamics is less affected than the longitudinal dynamics. It is worth noting that the SNR is decreased by 6.13% despite of the RMS of the slip ratio error being 0.0786 (nearly half the optimal slip ratio of 0.14), as seen in Figure 13d. The limited adjustment ranges for the tire and the bearing stiffness hinder further improvements of vehicle dynamics through parameter optimization.

The UEMF generated by the multi-field coupling effect exacerbates motor vibration, shortening the service life of motor and reducing the ride comfort of vehicle. Simulation results for UEMF are provided in Table 6, while the response comparisons are depicted in Figure 14.

**Table 6.** Optimization results of UEMF.

| RMS | Vertical UEMF (N) | Longitudinal UEMF (N) |
| --- | --- | --- |
| Before optimization | 117.95 | 171.01 |
| After optimization | 77.02 | 133.06 |
| Optimization results | 34.7% ↓ | 22.2% ↓ |

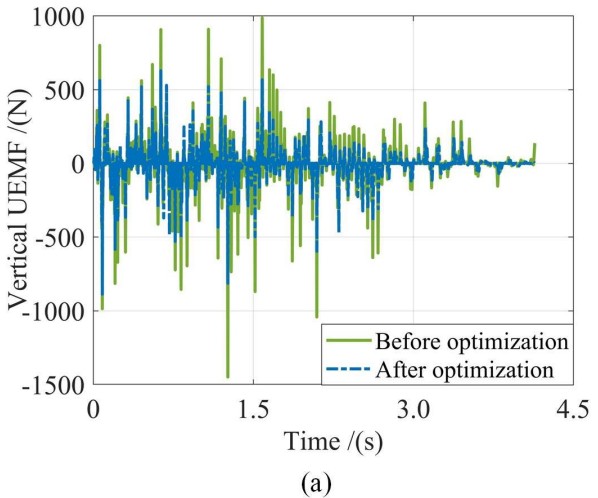
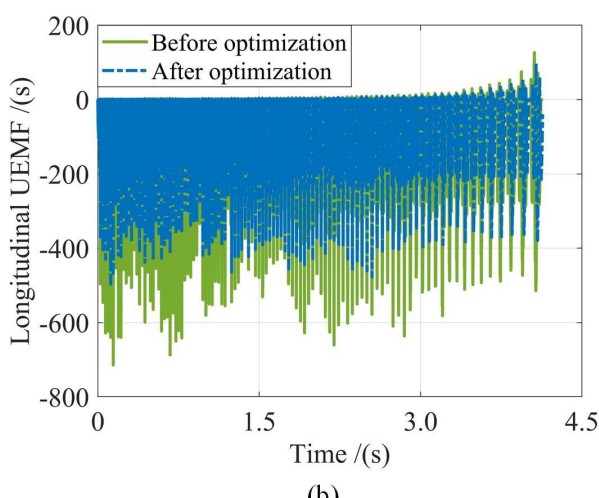

(a)

(b)

**Figure 14.** Optimization results of UEMF: (**a**) vertical UEMF and (**b**) longitudinal UEMF.

Generally, multi-objective optimization can improve the vertical–longitudinal dynamic performance of a vehicle to a certain extent. In addition, it is worth noting that the influence of UEMF can be effectively reduced by the TOPSIS algorithm.

## 5. Conclusions

In this paper, a complete and thorough investigation on the vertical–longitudinal coupling effects of in-wheel-motor-drive electric vehicles (IWMD EVs) is conducted. A comprehensive suspension-in-wheel-motor system (SIWMS) model is first established, in which a road–tire–ring force (RTRF) model is built to simultaneously capture the vertical and longitudinal vehicle dynamics. Further investigations are then conducted under different braking maneuvers with various road conditions to reveal the negative effects of road–tire–ring force (UEMF) on vehicle dynamics. Furthermore, a virtual prototype environment is developed to validate the efficacy of the developed SIWMS. Four key parameters are optimized using a multi-objective optimization method. The simulation results show that UEMF can significantly compromise longitudinal vehicle dynamics while slightly affecting vertical dynamics. Through optimization, the longitudinal acceleration rate and the signal-to-noise ratio of the slip ratio are respectively reduced by 5.07% and 6.13%, while the UEMF in the vertical and longitudinal directions are reduced to the ranges of from 22.2% to 34.7%, indicating improved the ride comfort and handling performance of vehicle.

**Author Contributions:** Data curation, Z.Z. and L.G.; Methodology, J.W. and S.L.; Project administration, J.W. and L.G.; Resources, L.Z. and J.W.; Software, S.L.; Supervision, L.Z. and L.G.; Validation, L.Z.; Visualization, Z.Z.; Writing—original draft, Z.Z. and L.Z.; Writing—review and editing, Z.Z. and L.Z. All authors have read and agreed to the published version of the manuscript.

**Funding:** This research was supported in part by the National Natural Science Foundation of China (Grant No. 52272387 and Grant No. 52202457) and in part by the State Key Laboratory of Mechanical Behavior and System Safety of Traffic Engineering Structures, Shijiazhuang Tiedao University (Grant KF2020-29).

**Institutional Review Board Statement:** Not applicable.

**Informed Consent Statement:** The data involved in this paper do not involve ethical issues.

**Data Availability Statement:** The datasets used and analyzed during the current study are available from the corresponding author on reasonable request.

**Conflicts of Interest:** All authors certify that they have no affiliations with or involvement in any organization or entity with any financial interest or non-financial interest in the subject matter or materials discussed in this manuscript.

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
