# Peer review of "Vertical-Longitudinal Coupling Effect Investigation and System Optimization for a Suspension-In-Wheel-Motor System in Electric Vehicle Applications"

_sustainability, doi:10.3390/su15054168_

Round 1

Reviewer 1 Report

Vertical-longitudinal coupling effect investigation and system optimization of a suspension in-wheel-motor system in electric vehicle applications

Ze Zhao, Lei Zhang, Liang Gu, Jianyang Wu

Vehicles with in-wheel motors, it's already a well-known solution. The authors present a comprehensive study to reveal the coupled vertical-longitudinal effect on suspension-in-wheel-motor system (SIWMS) and to present a viable optimization procedure to improve ride comfort and handling performance.

The research idea is interesting, and the manuscript is generally well structured.

I propose that the authors take into account the new references that can be found in., e.g.

Shuwen He, Xiaobin Fan, Quanwei Wang, Xinbo Chen, Shuaiwei Zhu, Review on Torque Distribution Scheme of Four-Wheel In-Wheel Motor Electric Vehicle, Machines 2022, 10(8), 619

Ślaski, G.; Gudra, A.; Borowicz, A. Analysis of the influence of additional unsprung mass of in‐wheel motors on the comfort and safety of a passenger car. Arch. Automot. Eng. 2014, 3, 51–64

Dukalski, P.; Będkowski, B.; Parczewski, K.; Wnęk, H.; Urbaś, A.; Augustynek, K. Dynamics of the vehicle rear suspension system with electric motors mounted in wheels. Maint. Reliab. 2019, 21, 125–136.

Dukalski, P.; Będkowski, B.; Parczewski, K.; Wnęk, H.; Urbaś, A.; Augustynek, K. Analysis of the Influence of Motors Installed in Passenger Car Wheels on the Torsion Beam of the Rear Axle Suspension, Energies 2022, 15(1), 222

Author Response

Reviewer: #1

Comments to the Author

  1. Vehicles with in-wheel motors, it's already a well-known solution. The authors present a comprehensive study to reveal the coupled vertical-longitudinal effect on suspension-in-wheel-motor system (SIWMS) and to present a viable optimization procedure to improve ride comfort and handling performance.

The research idea is interesting, and the manuscript is generally well structured.

Authors' Response:

Thank you very much for your precise summary of our work. We have revised our manuscript based on the comments from you and other reviewers. Thus, we resubmitted the revised manuscript for your further consideration. The specific revisions will be presented in the following parts.

  1. I propose that the authors take into account the new references that can be found in., e.g.

Authors' Response:

Thanks a lot for your advisable suggestion. As you suggested, we have refined the literature review to better account for the necessity of studying the vertical-longitudinal coupling effect on ride comfort for in-wheel-motor-drive electric vehicles (IWMD EVs). The new references you suggested have been included in the Introduction part. The relevant revisions have been highlighted and enclosed below for your easy reference.

Reviewer 2 Report

Please add practical implication of the findings and the topic in a more practical form within the conclusion and abstract.

Author Response

Reviewer: #2

Comments to the Author

Please add practical implication of the findings and the topic in a more practical form within the conclusion and abstract.

Authors' Response:

We are sincerely grateful to your effort in reviewing our manuscript. As you suggested, we have refined the Abstract and the Conclusion to better account for the necessity of studying the vertical-longitudinal coupling effect on ride comfort and handling performance for in-wheel-motor-drive electric vehicles (IWMD EVs). The relevant revisions have been highlighted in the revised manuscript.

Reviewer 3 Report

The paper presents an interesting idea of coupling longitudinal and vertical dynamics of the in-wheel motor EVs. To the reviewer's knowledge, this approach is new. The article is well structured.

Some comments regarding the paper are for clarification purposes only.

1) In Line 155, it is mentioned that a 5 kW electric motor is used. EVs use motors with much larger power. Does this model still work when rescaled to a larger power and weight?

2) In Figure 5, k_ux is indicated as shaft sleeve stiffness, which affects only longitudinal dynamics. Should not it influence the vertical dynamics as well?

3) Line 374 - "specify" should be corrected

4) Line 376 - what does "collision model" means? 

5) Table 4 - probably better to use the word "degradation"  rather than "worseness"

Author Response

Thank you very much for the professional review of our submission and for your valuable comments. We have revised our manuscript based on the comments. Thus, we submitted the revised manuscript for your further consideration. The specific revisions have been highlighted in the revised manuscript, and an independent response letter to address your comments is also enclosed. You may refer to the attached document for detailed responses to your comments. 
